# Neuronal Cell Rearrangement During Aging: Antioxidant Compounds as a Potential Therapeutic Approach

**DOI:** 10.3390/cells13231945

**Published:** 2024-11-23

**Authors:** Erjola Bej, Patrizia Cesare, Michele d’Angelo, Anna Rita Volpe, Vanessa Castelli

**Affiliations:** 1Department of Life, Health and Environmental Sciences, University of L’Aquila, 67100 L’Aquila, Italy; erjola.bej@graduate.univaq.it (E.B.); patrizia.cesare@univaq.it (P.C.); michele.dangelo@univaq.it (M.d.); 2Department of the Chemical-Toxicological and Pharmacological Evaluation of Drugs, Faculty of Pharmacy, Catholic University Our Lady of Good Counsel, 1001 Tirana, Albania

**Keywords:** neurodegeneration, brain, oxidative damage, antioxidants, ROS, mitochondrial dysfunction, organelles

## Abstract

Aging is a natural process that leads to time-related changes and a decrease in cognitive abilities, executive functions, and attention. In neuronal aging, brain cells struggle to respond to oxidative stress. The structure, function, and survival of neurons can be mediated by different pathways that are sensitive to oxidative stress and age-related low-energy states. Mitochondrial impairment is one of the most noticeable signs of brain aging. Damaged mitochondria are thought to be one of the main causes that feed the inflammation related to aging. Also, protein turnover is involved in age-related impairments. The brain, due to its high oxygen usage, is particularly susceptible to oxidative damage. This review explores the mechanisms underlying neuronal cell rearrangement during aging, focusing on morphological changes that contribute to cognitive decline and increased susceptibility to neurodegenerative diseases. Potential therapeutic approaches are discussed, including the use of antioxidants (e.g., Vitamin C, Vitamin E, glutathione, carotenoids, quercetin, resveratrol, and curcumin) to mitigate oxidative damage, enhance mitochondrial function, and maintain protein homeostasis. This comprehensive overview aims to provide insights into the cellular and molecular processes of neuronal aging and highlight promising therapeutic avenues to counteract age-related neuronal deterioration.

## 1. Introduction

Aging is a natural process that affects all organs, including the nervous system. Neurons, compared to other cell types, are more likely to change in function and metabolism due to insults, diseases, and/or neurodegenerative disorders, especially those that are age-related. In neuronal aging, brain cells find it difficult to respond to oxidative stress due to the accumulation of aggregated proteins in the cytoplasm and mitochondria [1]. Neurons tend to shrink and morphologically change the structure of their dendrites over time. According to studies conducted by Buell and Coleman, dendritic growth remains possible even in the aging human brain [2].

Aging is related to decreased executive function, cognitive abilities, and attention impairment [3]. Additionally, aging leads to a reduction in dopamine, serotonin (5-HT), and brain-derived neurotrophic factor (BDNF) levels, which regulate neuronal survival. Dopaminergic neurotransmitter systems are persistently adjusted during aging [4]. Compromised 5-HT and BDNF are central to the pathogenesis of neurodegenerative diseases such as Alzheimer’s disease [5]. On the other hand, there is an increment in monoamine oxidase, which leads to the catabolism of the neurotransmitters releasing hydrogen peroxide $(H_{2}O_{2}$) which can trigger the formation of hydroxyl radicals $O_{2}^{-}\cdot$ [6,7]. Parameters such as regional cerebral blood flow and regional cerebral metabolic rates for glucose and oxygen are indicators of normal brain activity, and they tend to decrease remarkably during aging [8]. Moreover, many neurotrophic factors and cytokines may contribute to neurogenesis after injury [9,10]. Dietary reductions, resulting in increased production of protein chaperones in brain cells, along with the use of folic acid, can improve cognitive abilities in mouse models of AD or PD [11]. Additionally, the use of antioxidants, such as saffron, can have neuroprotective effects by modulating neurotransmitters, reducing inflammation, oxidative stress, and apoptosis [12].

The first part of this review aims to provide insights into the cellular and molecular processes of neuronal aging, while the second part discusses promising therapeutic interventions based on antioxidant origins to counteract age-related neuronal deterioration.

## 2. Materials and Methods

The literature reviewed included preclinical and clinical studies. The in vitro and in vivo studies were selected based on the following points of view: neurological aging, mitochondrial dysfunction, oxidative damage, and antioxidants. Literature search was run by Google Scholar, PubMed, and free PMC articles using results up to August 2024.

The research queries included the following terms: brain aging, neuronal cell rearrangement, and proteostasis, combined with terms to determine the outcomes of interest such as neuroprotection, therapeutical approaches in aging, and antioxidants effects.

The figure was made using Photoshop software cc 2019 (version 20.0), and it is original.

From the literature search, we obtained more than 16,000 reviews, from which 221 studies were identified and analyzed, and 179 were taken in consideration as suitable for this article review.

## 3. Age-Related Alterations

### 3.1. Age-Related Changes in the Brain

Aging goes along with progressive neuron loss and cognitive and behavioral decline [13,14]. The cognitive deficits include not only memory loss but also decreased motor performance [15,16]. The aging process is related to impairment of the central nervous system [17]. Brain aging is also associated with a reduction in neurophysiological functions and neuro-inflammation, neuron cell shrinking, dendritic degradation, demyelination, and microglial activation [18]. One of the main characteristics of an aging brain is neuro-inflammation induced by the glial cells [19]. Cellular senescence can play a crucial role in the aging brain [20]. Studies in recent years, conducted on genetic mouse models, have revealed the effects of anti-inflammatory agents such as ibuprofen, which defend against cognitive decline induced by chronic inflammation by reducing markers of senescence in neurons [21]. Normally, starting from mid-life, the volume and the cortical thickness begin to decline with the passing of years. What we notice more frequently is that injuries such as white matter hyperintensities start showing with age [22]. In the first stages, the aging mechanisms include mostly the cells, but then, slowly, tissue and organ alterations begin to happen [23]. The age-dependent brain structure differences in the frontal lobe and medial temporal regions are related to changes in cognitive functions of the same regions. What is observed is also a relation between degeneration of age-related white matter segments and cognitive performance [24]. To achieve a particular target in an aging brain, the adaptive brain uses a normal mechanism called scaffolding, through which different and replacement neural circuits are used [25]. Neurofibrillary tangles (NFTs) and senile plaques (SPs) are found in physiological conditions during aging but also in pathological conditions such as Alzheimer’s disease (AD). The lesions tend to worsen with age and progression of the disease [26]. Neuronal loss might be the cause of cognitive disabilities and structural changes. In an aged brain, the most commonly reported are accumulation of Aβ protein and α-synuclein [27].

During aging, neurons experience morphological changes such as dendritic length, decrease in spine numbers, and as a result changes in synaptic densities. All the studies suggest that neuronal dysfunction in physiological age-related cases is due to changes inside the cortex resulting in myelin dystrophy and variations in synaptic transmission [28]. Intracellular accumulation of Aβ protein leads to oxidative stress and mitochondrial dysfunction, which may lead to neuronal death [29]. Structure, function, and survival of neurons can be mediated by different pathways that are all sensitive to oxidative stress and age-related low-energy states [26].

Heat shock proteins (HSPs), also called stress proteins, are cellular components, acting as molecular chaperones, whose main function is to control the folding of proteins; additionally, the HSPs are able to prevent the cellular aggregation of amyloid peptide and hyperphosphorylated tau in neurodegenerative diseases. Stress proteins participate in protein synthesis and transport, and they are activated in stressful situations. Being highly expressed in the nervous system, these proteins protect the neurons and glia [30]. They also regulate the apoptosis-controlling inflammatory pathway while secreting inflammatory mediators, assist in the degradation of abnormal proteins, and modulate antigen presentation. A collapse or malfunction of these proteins contributes to the generation of these insoluble hyperphosphorylated inclusions [31].

### 3.2. Mitochondrial Dysfunction in Brain Aging

Mitochondrial dysfunction is one of the most remarkable signs of brain aging [32]. According to recent studies, aging is the outcome of free radicals originally generated by cellular processes that are harmful, dangerous, and might lead to impairment [33]. Mitochondria are organelles exerting a crucial role in energy production, controlling the cell cycle from growth to apoptosis, regulating intracellular calcium, producing reactive oxygen species (ROS), and so on [34]. As we age, their efficiency in energy production declines, leading to reduced cellular function. Additionally, aging mitochondria generate more ROS, which can damage cellular components, including mitochondrial DNA (mtDNA). Over time, mutations accumulate in mtDNA, further impairing mitochondrial function and contributing to cellular aging. This dysfunction can also trigger cellular senescence, where cells stop dividing and secrete inflammatory factors, exacerbating tissue aging and age-related diseases. Also, dysfunctional mitochondria in many age-related diseases such as Alzheimer’s, Parkinson’s, and Huntington’s diseases have been reported, as well as in inflammatory conditions [35].

Specifically, one of the many main theories of aging indicates that oxidative stress, produced during the phosphorylation of mitochondrial macromolecules such as proteins, lipids, and mtDNA, is the one responsible for aging. We observe deletions of mtDNA more frequently in aged generations [36]. According to some studies, mtDNA deletions are more probable to happen during the adjustment of the injured mtDNA than during its replication [37]. It seems that ROS are the main factor responsible for mtDNA mutations during aging [38]. mtDNA is responsible for the encoding of crucial elements of oxidative phosphorylation and the synthesis of proteins [39]. Therefore, oxidative damage-induced mtDNA mutations that damage the function of the respiratory mitochondrial chain will cause the formation of more and more ROS, resulting in cell death [40,41]. Aging as a physiological process reduces the efficiency of oxidative phosphorylation, resulting in increased ROS production and altered ATP production. The great amount of ROS produced oxidizes more macromolecules eventually; consequently, we observe a worsening of mitochondrial function, generating more ROS in a vicious cycle [42]. Mitochondria play a crucial role in controlling the apoptosis activating the permeability transition pore. This activation is intensified in the brain of aged mice [43]. Aging is also connected to a reduced ability to produce ATP by oxidative phosphorylation in mammalian brains [44]. What we observe is a decrease in ATP production capacity with time [45]. At this point, cells should be either able to function with less energy or they should find a way to produce energy through non-oxidative metabolism. All this will affect other mechanisms, metabolisms, and signaling pathways [46].

The old mitochondria reduce the synthesis of mitochondrial proteins, lower the activity of oxidative enzymes, and minimize the mitochondrial mass, resulting in a decrease in the production of ATP [47]. The functions of ATP-dependent proteases, such as AFG3L2, AFG3L2/SPG7, LONP1, and YME1L, which regulate functions such as ribosome assembly, transcription and translation, adaptation to hypoxia, protein import, lipid trafficking, and mitochondrial dynamics, have been shown to lessen with aging, resulting in the inability to maintain a healthy and functional mitochondrion [48]. YME1L and OMA1 are proteases located in the inner membrane of the mitochondria responsible for the regulation of mitochondrial morphology [49]. They can degrade one another based on insults, leading to depolarization of the mitochondrial membrane and consumption of ATP; as a result, they can balance mitochondrial dynamics [50]. LONP1, on the other hand, supports protein folding while arranging the proteolytic activity [51].

During aging, senescent cells tend to accumulate, causing a functional decline leading to morbidity and mortality [52]. Mitochondrial dysfunction in senescent cells can be considered as both a reduction in respiratory function and a decreased mitochondrial membrane potential (MMP). We can benefit from having the MMP decrease at a steady rate because the electron transport chain (ETC) passes in a more oxidized state, reducing ROS production, but also the ETC can become non-functional [53]. In elderly people, the respiratory chain complexes form super complexes, which, if not stable, lead to the formation of more ROS [54,55].

To maintain MMP and to produce ATP, it is essential for mitochondria to find the balance between $NAD^{+}$ and NADH, as this ratio is very important for mitochondrial metabolism. The levels of $NAD^{+}$ within the cells are regulated by a balance between its production and its degradation [56]. $NAD^{+}$ consumption during stress and aging and the disruption in $NAD^{+}$/NADH will lead to mitochondrial dysfunction, mitochondrial membrane depolarization, and altered mitochondrial permeability transition, which are connected to cell death [57].

Inflammation is a physiological process that arises after an insult with the main aim of restoring cells and damaged tissues after eliminating the danger. After this phase, the inflammatory process should come to an end; if not, the inflammation might become chronic. Non-functional mitochondria are thought to be one of the main causes that nourish inflammation related to aging. Respiratory dysfunction and oxidative stress are the mechanisms through which the mitochondria get involved in the aging process [58]. To avoid the release of the content of the damaged or aged mitochondria in the cytoplasm or extracellular space, it is very important to sequester the defective mitochondria in a double-membraned autophagosome via mitophagy, with the consequence of having a minor risk of developing chronic inflammation [59]. Impaired age-related mitochondria can also unbalance calcium buffering, leading to a variation in the concentration of free calcium ions in cells [60], causing obstacles with synaptic activity since $Ca^{2+}$ participates in the transmission of the signal. $Ca^{2+}$ homeostasis can be regulated by protein channels, which we find in both inner and outer mitochondrial membranes and by the endoplasmic reticulum [61]. During aging, neurons find it hard to maintain the required energy level, and this influences cytoplasmic $Ca^{2+}$ levels. Therefore, we find $Ca^{2+}$ signaling to be disturbed and associated with a debilitating impairment [62]. $Ca^{2+}$ increments in the cytosol force the mitochondria to rapidly uptake it to avoid overloading the cytosol. This, on the other hand, can result in mitochondrial overcharge, ROS generation, and ATP depletion [63].

### 3.3. Morphological Rearrangement in Brain Aging

Different studies have shown that interruptions in brain synaptic circuitry can contribute to cognitive decline [64]. During physiological aging, there is a decline in the white matter in the cerebral hemispheres and structural irregularities in myelin [65]. With age, the structure of myelin undergoes alterations, and the axons start showing evidence of degeneration, resulting in the loss of nerve fibers. Additionally, according to a study by Sandell conducted on monkeys, the number of myelinated nerve fibers tends to decrease over time compared to young monkeys [66]. The structure of the myelin sheath undergoes age-related changes and modifications, leading to defective and weak conduction speed of neuronal signals along the axons [67].

Protein turnover in the brain is a process that undergoes age-related impairments. In addition, altered proteolysis of various cell types is believed to be related to cytotoxicity. Protein turnover is thought to be related to protein oxidation and aggregation of different forms, such as Lewy bodies and lipofuscin granules [68]. Accumulation of lipofuscin granules is found in both senescence and various age-related and neurodegenerative diseases such as Alzheimer’s disease. Lipofuscin can accumulate because cells find it hard to exocytose it [69]. Neurons that contain a great quantity of lipofuscin under physiological conditions are less affected by degenerative alterations [70]. Forming lipofuscin is understood as a form of neuronal adaptation, and those neurons that lack this capability are more vulnerable to degeneration [70,71,72]. Nevertheless, evidence shows that accumulated lipofuscin can be dangerous, interfering with normal cellular functioning because its deposits contain toxic compounds [73]. Therefore, lipofuscin elimination can be a possible target in anti-aging therapies [74]. The study conducted on rats while being treated with melatonin or coenzyme Q10 for 4 weeks revealed a reduction in lipofuscin [75]. The presence of lipofuscin increases with age [76].

During aging, carbohydrate-rich inclusions in the neuronal soma are found, also called glycogen inclusions or polyglucosan bodies, which can also contain proteins. These inclusions are both found in aging brains and in neurodegenerative diseases and might play a powerful role in cell cycles. However, it is not yet clear whether these carbohydrate bodies are the cause or the result of brain senescence [77].

Axons tend to increase in diameter, becoming larger and thicker with age. Evidence has demonstrated that mitochondria generate more oxidative stress in axons, leading to a vulnerability to further injury because the mitochondria themselves suffer compromised functional integrity [78]. The selective destruction of axons is a process that occurs due to aging, and it is a characteristic of neurodegeneration because it induces neuronal deterioration [79]. The first process occurring is axonal degeneration; after that, neuronal death occurs [80]. Myelin degradation is subsequent to the breakdown of the axonal cytoskeleton [81]. This self-destructive process, with its molecular mechanism, can be an important target for the strategies implicated in neurodegenerative disorders. According to Lasiene, internode lengths undergo a significant reduction with age [82]. Axonal active destruction relies on the activation of the mitochondrial permeability transition pore (mPTP) between both mitochondrial membranes, the inner and the outer ones [83]. This activation, while provoking the mitochondrial permeability transition, increases the ROS in axons contributing to axonal degeneration [84].

### 3.4. Protein Degradation in Brain Aging

Protein degradation is a crucial process that helps break down and remove the aggregated or the misfolded ones [85]. Protein homeostasis, also called proteostasis, promotes healthy protein synthesis, folding, and degradation. In the meantime, biological molecules that can modify the conformation or the structure of proteins can be used as a potential target to improve the symptoms of some neurodegenerative disorders [86]. Additionally, small molecules considered to be pharmacological chaperones can stimulate the refolding of impaired proteins [87]. Proteolytic enzymes, such as peptidases/proteases, are found in different intracellular and extracellular compartments, and they can degrade proteins or in some cases form a fusion with the pharmacological chaperones to prevent non-specific aggregation [49]. The synthesis of the chaperones is a stress-induced process that is damaged and weakened during longevity [88]. Aging is connected to the misregulation of protein maintenance [89] and to the alteration of proteostasis [90]. The mechanism of proteostasis includes processes that lead to the balance of accurately folded proteins in their three-dimensional structure and processes where the proteasome or lysosome degrades the proteins [91]. Proteolytic systems, autophagy–lysosomal and ubiquitin–proteasome more specifically, are also age-dependent systems [92]. The ubiquitin–proteasome system is the one that regulates proteostasis in the nucleus [49]. Additionally, the ubiquitin proteases also regulate stem cell functions since they tend to decrease with the passing of the years [93]. Proteases can also operate as signaling molecules since we find them in both intracellular compartments and extracellular areas [49]. Moreover, they can have an impact on cell senescence since this last one is a causal factor of aging and can guide cellular reprogramming [49]. Membrane proteases along with extracellular proteases can regulate extracellular proteolysis [94]. Extracellular proteases themselves eliminate misfolded proteins because the unfolded protein response can, in an indirect way, modulate extracellular proteostasis [95]. Secreted extracellular chaperones, which act as intermediaries for misfolded proteins, help maintain proteostasis [96].

### 3.5. Oxidative Stress in Brain Aging

The brain, since it uses a great quantity of oxygen for energy production, is one of the organs that is at greater risk of oxidative damage. “Inflammaging” is a term used to describe the inflammation process generated by ROS and how this might affect the nervous system during aging [97]. “Inflammaging”, compared to inflammation, is a situation where the inflammation happens without having an apparent pathogen [98]. The main focus is not to eliminate the inflammation but to support the equilibrium between inflammaging and processes that contrast it [99]. One of the first steps of aging is an increased number of senescent cells [100] that accumulate in tissues while releasing pro-inflammatory cytokines and proteases [101]. Among the many theories of aging, the free radical theory is one of the most studied [102]. According to this theory, elderly people have excessive amounts of oxidized products compared to younger generations. Therefore, using antioxidants in this group is essential for increasing lifespan [103] (Figure 1). Lifespan is inversely proportional to the concentration of ROS in the mitochondria [104]. ROS are produced in normal cellular metabolism. The three most crucial ROS are the superoxide anion ${\;(O}_{2}^{-}\cdot$), the hydroxyl radical $\left(OH\cdot \right)$, and hydrogen peroxide ${(H}_{2}O_{2}$) [105] (Figure 1). The hydroxyl radical is the most reactive species, and it can be the cause of damage to all macromolecules such as lipids, proteins, carbohydrates, and nucleic acids. Defects or deficiencies of enzymes that are capable of scavenging free radicals such as super oxide dismutases can be correlated to the accumulation of free radicals [106]. Organic compounds are thermodynamically unbalanced in oxygen surrounding the generation of oxygen radicals, which might be the cause of damaging oxidative reactions [107].

## 4. Therapeutic Approaches to Counteract Brain Aging

There are several therapeutic approaches aimed at counteracting brain aging and neuronal cell rearrangement. As we mentioned above, mitochondrial dysfunction is closely associated with brain aging due to oxidative damage. Strategies to protect mitochondria include the use of antioxidants to reduce oxidative stress and the development of drugs that enhance mitochondrial function. Antioxidants play a crucial role in compensating for the negative effects of the ROS and scavenge free radicals while multiplying the antioxidant defense [108]. The antioxidant properties of different compounds are mainly supplied by phenolic acids, phenolic diterpenes, flavonoids, and volatile oils [109,110]. By their use, what we obtain is a dynamic but fine-tuned redox equilibrium. Additionally, these molecules can also prevent or delay the oxidation of substances that cause the formation of ROS and interrupt propagation of autoxidation [111]. The antioxidant action of these compounds is probably connected to their capability to adjust the activity of the redox enzymes while interconnecting with redox signaling pathways [112]. The activity of the antioxidants in vivo is multiplied because those that cannot be absorbed in the small intestine can be split into smaller molecules, not only increasing the possibility of better absorption but also resulting in many different potential targets [113]. Many dietary antioxidants get involved in different redox regulations, such as MAPK-mediated pathways that are responsible for controlling the expression of many genes. Polyphenols, while activating Nrf2, can enhance the activity of antioxidant enzymes like CAT, GPx, glutathione reductase, and paraoxonases [114]. Through these redox-balanced mechanisms, we obtain lifespan-increasing effects [115]. Another mechanism through which antioxidants can manifest their effects is by interacting with specific proteins which play a crucial role in intracellular signaling [116]. Playing a role as a reducing agent and interfering with metal ion chelation while eliminating free radicals are ways through which antioxidants can exert their effect [117]. When the protection that comes from endogenous antioxidants is not enough, we might rely on the help that we find from diet- or drug-derived antioxidants [118].

In this review, we will focus on emerging evidence about Vitamin C, Vitamin E, glutathione, carotenoids, quercetin, resveratrol, and curcumin intake during aging.

Vitamin C, also known as ascorbic acid, plays an antioxidant role by scavenging oxygen free radicals [119] (Table 1). The concentration of ascorbic acid in the body is considered to be the highest compared to other types of vitamins [120]. An absence of Vitamin C in the brain of mouse models, affected by Alzheimer’s disease, indicates that there is an imbalance in the redox processes stimulating the generation of β-amyloid [121]. According to the study conducted by El-Gendy et al., Vitamin C might be able to reduce the oxidative damage caused by imidacloprid. The protective effect of the pre-treatment is better compared to the post-treatment [122] (Table 1). One of the influencing factors of healthy aging is the daily intake of Vitamin C, reducing the risk of mortality [123] (Table 1). As specified by Brubacher et al., a lower mean Vitamin C status appears in a group of older people compared to a group of young ones after an intake of 60 mg/day. Thus, the demand for Vitamin C in the older generations is more elevated [124]. On the other hand, the study by Murata and colleagues compared healthy and young adults with older adult groups, who were chronically ill and hospitalized. The group of elderly people following long-term supplements showed decreased Vitamin C status [125].

Vitamin C is not the only one to have a lower concentration in AD patients; Vitamin E also shows reduced concentration in contrast to subjects with normal cognitive functions [126]. Vitamin E, also known as α-tocopherol, is concentrated mostly in the interior part, the hydrophobic one, of the cell membrane and plays a role as a shelter in case of membrane injury. Additionally, Vitamin E slows down and interferes with the production of ROS and stimulates the apoptosis of cancer cells [105] (Table 1). Vitamin E is an antioxidant that acts also as a neuroprotective agent and can be used to prevent neurodegenerative diseases [127,128] (Table 1). Different studies revealed that the combination of Vitamin C and Vitamin E prevents and blocks age-associated impairments of cognitive abilities [129]. In the meantime, the studies conducted on old rats, while combining both vitamins, show a decrease in the oxidative stress and protection of the hypothalamus and cortex in case of cold exposure-induced oxidative stress resulting in better protection against oxidative stress [130,131,132].

Glutathione (GSH) is the most soluble antioxidant. Glutathione purifies hydrogen peroxide and preserves the membrane lipids from oxidative stress. In addition, GSH is able to keep the cells safe while working alongside proapoptotic and antiapoptotic signaling pathways [133] (Table 1). GSH is a tripeptide containing cysteine, glutamic acid, and glycine, found mostly in the glial cells, and it is necessary for cell proliferation and ROS neutralization [134] (Table 1). Additionally, GSH is related to the impediment of mitochondrial damage and cell death. GSH might function as a signaling molecule in the CNS as well [135]. During aging and neurodegenerative diseases, what we observe is a debilitation of GSH functions, and this is due to a loss of neurons [136]. At this elderly stage of life, GSH plays a very important role in defending neurons in case of damage caused by ROS [137] (Table 1). An age-dependent reduction in GSH levels can be the cause of the outbreak of age-dependent neurodegenerative disorders [138,139].

Carotenoids such as β-carotene tend to react with oxygen radicals. The antioxidant effect of carotenoids is mainly demonstrated in low oxygen partial pressure [140]. In patients suffering from neurodegenerative diseases, especially those affected by AD, the blood concentrations of carotenoids were identified to be low [141]. One of the mechanisms through which β-carotene performs its antioxidant property is the interaction with biological membranes [142] (Table 1). Studies demonstrate that (Z)-isomers of β-carotene have antioxidant activity in vitro [143]. The β-carotene activity should also be seen from another point of view. As we all know, a telomere’s length shortens with age, and that might result in apoptosis and senescence. It looks like, according to the study by Boccardi et al., that β-carotene can adjust the telomerase activity while aging, protecting the telomers from oxidative stress [141] (Table 1). β-carotene is an antioxidant with a potential association to the maintenance of cognitive functions, and apparently it works better in synergy with other supplementations such as Vitamin E, Vitamin C, zinc, and selenium [144]. This was proved by a study conducted on men treated with 50 mg of β-carotene for 18 years. Their cognitive memory was better compared to the men treated with a placebo. The benefits can be attributed to the early age of starting to consume it and/or the long period of using it [145]. β-carotene along with Vitamin E can prevent lipid peroxidation [146]. In animal models, increased levels of β-carotene are associated with higher levels of the neurotrophin brain-derived neurotrophic factor (BDNF), which has a crucial role in synaptic plasticity [147].

Quercetin is a flavonoid that is able to contrast the age-related increase in oxidative stress in the mitochondria, ameliorating the mitochondrial dynamics [148] (Table 1). It is also capable of slowing down cellular aging, boosting cell proliferation [149] (Table 1). Quercetin is not only an antioxidant, but it is involved in antiapoptotic and anti-inflammatory processes as well, so it can be thought of as a possible nutraceutical to be used in age-related disorders [150]. The accumulation of free radicals, while increasing oxidative stress, contributes to the production and release of pro-inflammatory cytokines such as interleukin 1β, tumor necrosis factor, and interferon γ [151]. Sirtuin 1 (SIRT1), which is a member of the NAD+-dependent deacetylase enzyme family, while adjusting the expression of pro-inflammatory cytokines, can diminish the degeneration of dopaminergic neurons [152]. Therefore, SIRT1 may be a promising therapeutical agent against age-related diseases [153]. Quercetin stimulates the activity of SIRT1, modulating the functions of inflammatory cytokines and therefore reducing ROS production and neuronal cell death [150,154] (Table 1). In the study conducted on aged mice, quercetin dampened neuro-inflammation through the SIRT1/NLRP3 pathway [155].

Several studies, based on cognitive functions revealed that the polyphenol resveratrol can protect cells from neuronal damage [156]. According to studies by Caldeira et al., resveratrol’s antioxidant mechanisms change depending on how old the cell is. The study by Caldeira conducted on the human mononuclear cells of two different kinds of donors, middle-aged and elderly donors, reflects how there is an increment of the activity of the superoxide dismutase enzyme in the cells of the elderly group after being exposed to resveratrol, the group where the highest levels of ROS production were also found [157]. Mitochondrial function can be preserved using resveratrol [158] (Table 1). The brain can benefit from resveratrol treatments because they may result in improved memory while reducing oxidative damage [159,160] (Table 1). Resveratrol’s main functions, based on anti-aging mechanisms, are enhancing oxidative stress while improving mitochondrial function and regulating cell apoptosis; therefore, it could be a bioactive compound to be used to target age-dependent diseases [161]. Resveratrol is considered a powerful antioxidant because it can not only clean free radicals, but it can also influence the increase in the activity of other antioxidant enzymes [162] (Table 1). While acting as a purifier of ROS, resveratrol can prevent lipid peroxidation [163] (Table 1). The higher the concentration of this polyphenol, the more antioxidant properties it reveals [164]. In cases of neurological disorders, resveratrol can inhibit the polymerization of the β-amyloid peptide [165]. Since resveratrol, after entering the bloodstream, can pass the blood–brain barrier, it can be used as a possible compound targeting neurodegenerative diseases [166,167,168]. According to the study by Fiorillo et al., resveratrol inhibits the mitochondrial ATP synthase acting as an anti-aging compound [169].

Among the antioxidants, we also mentioned curcumin, whose nanomicelles can block mitochondrial dysfunction related to brain aging [170] (Table 1). The effects of curcumin are also dependent on cell age, and this is proved by administering it to middle-aged monkeys for a period from 14 to 18 months. The results of the study conducted on middle-aged monkeys, whose cognitive decline had just begun, showed that curcumin can enhance cognitive performance. However, more research is needed in older monkeys, who also present a rise in inflammatory markers and serious cognitive disabilities [171,172]. Furthermore, curcumin can be defensive towards rat hippocampus cells in case of homocysteine (Hcy) neurotoxicity to enhance memory deficiency [173]. The clinical trials demonstrated that since curcumin can be effective in reducing oxidative stress and inflammation, this Indian spice can be used in cases of age-related cognitive decline [174] (Table 1). In the in vivo studies using neonatal Sprague-Dawley rats, where curcumin was administered at a dose of 100 mg/kg, we observed not only benefits in the improvement of white matter injury but an inhibition of the expression of iNOS and NOX in the microglia as well [175]. The fact that the nanoformulation of curcumin can increase NF-kB expression in fibroblast cells should encourage more studies to use curcumin as an anti-aging formulation [176,177]. Curcumin can exert a protective role during inflammaging by blocking cognitive malfunction in transgenic AD mice [178,179].

**Table 1 cells-13-01945-t001:** Main antioxidant compounds acting on brain aging.

Antioxidants	Benefits	References
Vitamin C 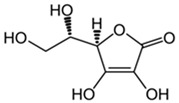	Scavenges oxygen free radicals. Reduces the oxidative damage caused by imidacloprid. Reduces the risk of mortality.	[119,122,123]
Vitamin E 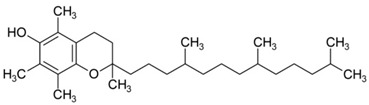	Plays a role as a shelter in case of membrane injury. Slows down the production of ROS. Prevents neurodegenerative diseases.	[105,127,128]
Glutathione 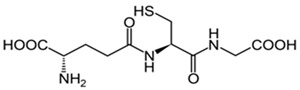	Preserves the membrane lipids from oxidative stress. Helps with ROS neutralization. Protect neurons from damage caused by ROS.	[133,134,137]
Β-Carotene 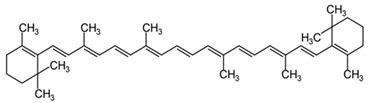	Interacts with the biological membranes, protecting them from oxidative damage. Modulates telomerase activity.	[141,142]
Quercetin 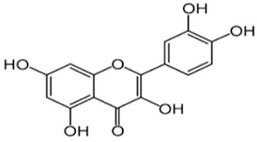	Contrasts the age-related increase in oxidative stress in mitochondria. Boosts cell proliferation. Stimulates SIRT1, reducing the ROS production and neuronal cell death.	[148,149,150,154]
Resveratrol 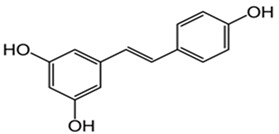	Helps to preserve mitochondrial function. Improves memory while reducing oxidative damage. Influences the activity of other antioxidant enzymes. Prevents lipid peroxidation.	[158,159,160,162,163]
Curcumin 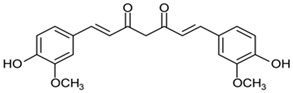	Blocks age-related mitochondrial dysfunction. Reduces oxidative stress and inflammation.	[170,174]

## 5. Discussion

Neuronal cell rearrangement during aging is a complex process influenced by various factors, including metabolism, oxidative stress, and mitochondrial function. As neurons age, they undergo significant changes that can lead to cognitive decline and increased susceptibility to neurodegenerative diseases.

Aged neurons often exhibit glucose hypometabolism, which affects their energy supply and can lead to impaired synaptic function and reduced neuronal plasticity. Increased oxidative stress is another hallmark of aging neurons, where the accumulation of ROS can damage cellular components, including DNA, proteins, and lipids. This oxidative damage is intensified in neurodegenerative diseases. Mitochondria, the powerhouses of the cell, become less efficient with age, contributing to reduced ATP production and increased ROS generation, further impairing neuronal function. Additionally, aging neurons often accumulate misfolded and aggregated proteins, which can disrupt cellular processes and contribute to neurodegenerative conditions like Alzheimer’s and Parkinson’s diseases.

Mitochondrial dysfunction is one of the most remarkable signs of brain aging. Aging is the outcome of free radicals originally generated by cellular processes that are harmful, dangerous, and might lead to impairment. We find dysfunctional mitochondria in many age-related diseases such as Alzheimer’s, Parkinson’s, and Huntington’s neurodegenerative diseases but also in inflammatory conditions. Free radicals can damage lipids, proteins, and carbohydrates. Peroxisomes are also organelles that go through morphological and structural age-related changes. Proteostasis promotes healthy protein synthesis, folding, and degradation, and this process is related to misregulation of protein maintenance during aging.

## 6. Conclusions

Aging is a natural and universal process that leads to time-related changes and a decrease in cognitive abilities, executive functions, and attention impairment. In neuronal aging, brain cells find difficulty in responding to oxidative stress due to accumulation of aggregated proteins in cytoplasm and mitochondria. Potential therapeutic approaches include antioxidants, which aim to reduce oxidative stress. The use of antioxidants can have neuroprotective effects while modulating neurotransmitters and reducing inflammation, oxidative stress, and apoptosis. Intracellular accumulation of Aβ protein leads to oxidative stress and mitochondrial dysfunction, which may lead to neuronal death. Structure, function, and survival of neurons can be mediated by different pathways that are all sensitive to oxidative stress and age-related low-energy states. Agents that protect mitochondrial function, such as mitochondrial-targeted antioxidants and peptides, can help maintain ATP production and reduce oxidative stress. Since most of the damage is caused by oxidative stress and ROS, the use of antioxidants such as Vitamin C, Vitamin E, glutathione, carotenoids, quercetin, resveratrol, and curcumin is essential in increasing lifespan. Furthermore, antioxidants play an important role in the prevention of mitochondrial damage and cell death, boosting cell proliferation and improving memory.

## 7. Limitations and Future Studies

Overall, neuronal cell rearrangement during aging is a multifaceted process that significantly impacts brain function. Understanding the underlying mechanisms can help develop targeted therapeutic strategies to mitigate age-related neuronal decline and improve quality of life for the elderly. To test the antioxidants’ efficiency further, clinical studies should be conducted. Also, it would be interesting to test different dosages of antioxidants, alone or in combination, to better understand the antioxidants’ effects and the scavenging free oxygen radicals’ properties. Future research should focus on combining pharmacological and lifestyle interventions to create comprehensive treatment plans for aging-related neuronal disorders.

## Figures and Tables

**Figure 1 cells-13-01945-f001:**
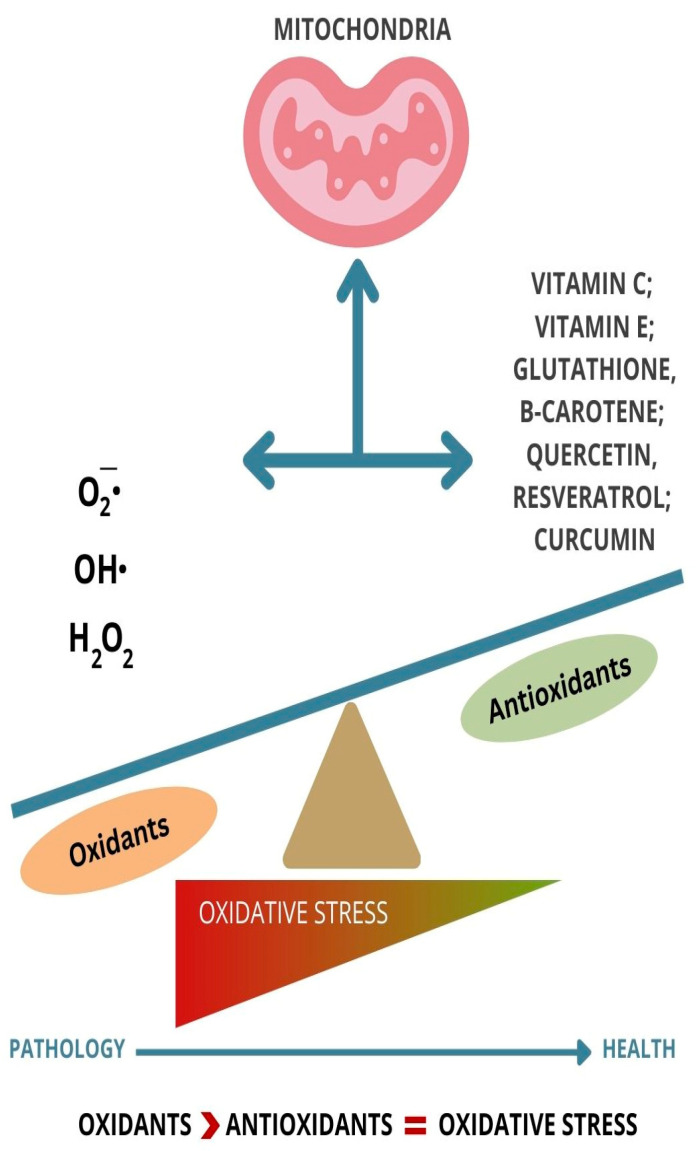
Balance and imbalance between ROS and antioxidants.

## Data Availability

Not applicable.

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
