# Peer review of "Neuronal Cell Rearrangement During Aging: Antioxidant Compounds as a Potential Therapeutic Approach"

_cells, 2024, doi:10.3390/cells13231945_

Round 1

Reviewer 1 Report

Comments and Suggestions for Authors

In the article entitled “Neuronal cell rearrangement during aging: potential therapeutic approaches” the authors review the mechanisms of neuronal cell rearrangement during aging, highlighting oxidative stress and mitochondrial deterioration as contributors to cognitive impairment and neurodegenerative diseases. They also discuss a therapeutic approach to counteract brain aging, particularly in the use of antioxidants such as vitamin C, vitamin E, as well as resveratrol to mitigate oxidative damage and improve mitochondrial function.

Some points that need to be addressed.

1.     In section 4 “Therapeutic approaches to counteract brain aging”, it would be important for the authors to address in more depth the mechanisms of therapeutic agents.

2.     Consult primary sources for therapeutic agents.

3.     In Table 1 in the reference section, cite with numbers.

Author Response

1st reviewer

In the article entitled “Neuronal cell rearrangement during aging: potential therapeutic approaches” the authors review the mechanisms of neuronal cell rearrangement during aging, highlighting oxidative stress and mitochondrial deterioration as contributors to cognitive impairment and neurodegenerative diseases. They also discuss a therapeutic approach to counteract brain aging, particularly in the use of antioxidants such as vitamin C, vitamin E, as well as resveratrol to mitigate oxidative damage and improve mitochondrial function.

Response: We would like to thank the Reviewer for the time spent revising our manuscript and for the positive comments. We tried to address all the points raised.

Some points that need to be addressed.

  1. In section 4 “Therapeutic approaches to counteract brain aging”, it would be important for the authors to address in more depth the mechanisms of therapeutic agents.

Response: We appreciate the Reviewer’s comments, and we modified part of the manuscript and the title as suggested also by other Reviewers.

  1. Consult primary sources for therapeutic agents.

Response: We thank the reviewer for the comment and modified the manuscript as suggested.

  1. In Table 1 in the reference section, cite with numbers.

Response: We apologize for the oversight and modified the table.

Reviewer 2 Report

Comments and Suggestions for Authors

The review aims to present changes in neuronal cells with the progress of aging. It is valuable scientific contribution which deserves to be published in Cells journal. The subject of review is interesting, broad and complicated. To make the review more legible for the reader it would be important for the review to present the subject in the form of scheme. Such scheme could more clearly introduce connections between the factors influencing the changes occurring with aging. The second problem is the title of manuscript. It does not reflect its content. It promises very broad treatment of the theme and it is not the case. “..Potential therapeutic approaches..” are limited to several antioxidants, coming from the natural sources. To present real subject of the review the title should be changed

Author Response

2nd Reviewer

The review aims to present changes in neuronal cells with the progress of aging. It is valuable scientific contribution which deserves to be published in Cells journal. The subject of review is interesting, broad and complicated.

Response: We would like to thank the Reviewer for the time spent revising our manuscript and for the positive comments. We tried to address all the points raised.

To make the review more legible for the reader it would be important for the review to present the subject in the form of scheme. Such scheme could more clearly introduce connections between the factors influencing the changes occurring with aging. The second problem is the title of manuscript. It does not reflect its content. It promises very broad treatment of the theme and it is not the case. “..Potential therapeutic approaches..” are limited to several antioxidants, coming from the natural sources. To present real subject of the review the title should be changed.

Response: We appreciate the Reviewer’s comments and we totally agree. We ameliorated the readability and modified the title as suggested.

Reviewer 3 Report

Comments and Suggestions for Authors

Please modify the title to correspond to the literature review and the methodology. “neurological aging, mitochondrial dysfunction, oxidative damage, and antioxidants.”

Pertinent words are mitochondrial dysfunction, oxidative damage, and antioxidants.

A figure summarizing the main idea of the manuscript is highly recommended. And this imaging should include oxidant and antioxidants because they are the main idea of the manuscript.

Could the authors explain the following statement: “graphic software, Canva, and AutoCAD?” What were the figures done with Canvas and AutoCAD? The current table and figures inside the table do not require specific software to be developed.

The authors should also explain better why they only found 212 studies based on their initial search. On October 22, 2024, the reviewer searched for aging and mitochondria and found 16,859 results.

https://pubmed.ncbi.nlm.nih.gov/?term=%28aging%29+AND+%28mitochondria%29&sort=date&ac=no

Please place the table soon after the first mention of the table.

Also, it is advised to review the text regarding grammar. There are extra spaces and points.

A specific chapter about limitations and future studies should be provided.

The conclusion should have a specific chapter.

Please review the references and read the instructions for authors. The current manuscript contains different types of references.

Author Response

3rd  reviewer

Please modify the title to correspond to the literature review and the methodology. “neurological aging, mitochondrial dysfunction, oxidative damage, and antioxidants. Pertinent words are mitochondrial dysfunction, oxidative damage, and antioxidants.

Response: We would like to thank the Reviewer for the time spent revising our manuscript and for the positive comments. We tried to address all the points raised. We agree with the  reviewer and modified the title as suggested.

A figure summarizing the main idea of the manuscript is highly recommended. And this imaging should include oxidant and antioxidants because they are the main idea of the manuscript.

Response: We thank the reviewer for the suggestion, we now included the figure.

Could the authors explain the following statement: “graphic software, Canva, and AutoCAD?” What were the figures done with Canvas and AutoCAD? The current table and figures inside the table do not require specific software to be developed.

Response: We apologize for the oversight; it is a misprint present in previous drafts.

The authors should also explain better why they only found 212 studies based on their initial search. On October 22, 2024, the reviewer searched for aging and mitochondria and found 16,859 results.

          https://pubmed.ncbi.nlm.nih.gov/?term=%28aging%29+AND+%28mitochondria%29&sort=date&ac=

          Response: We thank the reviewer and we clarified this point.

Please place the table soon after the first mention of the table.

Response: We moved after the first mention.

Also, it is advised to review the text regarding grammar. There are extra spaces and points.

Response: We thoroughly checked the whole manuscript for English grammar and typos using Ginger software and with the help of a native speaker as suggested.

A specific chapter about limitations and future studies should be provided.

Response: We totally agree with the reviewer and we now added this chapter as suggested.

The conclusion should have a specific chapter.

Response: We modified accordingly

Please review the references and read the instructions for authors. The current manuscript contains different types of references.

Response: We thank the reviewer and we used Zotero software to format them.

Round 2

Reviewer 1 Report

Comments and Suggestions for Authors

Thanks to the authors for their replies.

The manuscript improved considerably with the changes provided.

Reviewer 2 Report

Comments and Suggestions for Authors

After revision this work is suitable to publish in Cells journal